# Interventions Effective in Decreasing Burden in Caregivers of Persons with Dementia: A Meta-Analysis

**Francisco José Rodríguez-Alcázar** [1], **Raúl Juárez-Vela** [2], **Juan Luis Sánchez-González** [3,*]
**and Javier Martín-Vallejo** [4]

1 Regional Health Management of Castilla y León, 47007 Valladolid, Spain; obifran@hotmail.com
2 Group of Research in Care (GRUPAC), Faculty of Health Sciences, University of La Rioja,
26006 Logroño, Spain; raul.juarez@unirioja.es
3 Department of Physiotherapy, Faculty of Nursing and Physiotherapy, University of Salamanca,
37007 Salamanca, Spain
4 Department of Statistics, Faculty of Medicine; University of Salamanca, 37007 Salamanca, Spain; jmv@usal.es
* Correspondence: juanluissanchez@usal.es

**Abstract:** *Introduction*: Chronic non-communicable diseases, including diseases of mental origin such as Alzheimer's, affect all age groups and countries. These diseases have a major impact on the patient and their family environment. It is interesting that different questionnaires are measured in the same direction, given that different health questionnaires are used to measure caregiver burden. *Objectives*: To identify which type of intervention is the most appropriate to improve the health of the primary caregiver in patients with dementia. To understand the role played by the nurse within multidisciplinary teams and to know whether the different questionnaires used in the studies measure caregiver health in the same direction. *Methods*: A systematic search of the published and gray literature was carried out without restriction of the language used in the studies. Caregiver burden of patients with dementia, receiving an intervention to improve caregiver burden, was assessed. Standardized mean difference was used as the effect size measure, and there were possible causes of heterogeneity in the effect size. *Results*: In total, 1512 records were found, and 39 articles with 4715 participants were included. We found individual information with an effect of 0.48 (CI95%: 0.18; 0.79; I2 = 0%); group therapy with an effect of 0.20 (CI95%: 0.08; 0.31; I2 = 6%); workshops with an effect of 0.21 (CI95%: 0.01; I2 = 48%) and 0.32 (CI95%: 0.01; 0.54; I2 = 0%) when a nurse intervenes; respite care with an effect of 0.22 (CI95%: 0.05; 0.40; I2 = 66%); individual therapy with an effect of 0.28 (CI95%: 0.15; 0.4; I2 = 68%); and support groups with an effect of 0.07 (CI95%: 0; 0.15; I2 = 78%). *Conclusions*: The magnitude of the effects of the interventions has been low–moderate. Different instruments are not associated with the magnitude of the effect. The presence of nurses improves the effect of the intervention on caregivers when it is carried out in the form of workshops.

**Keywords:** dementia; caregiver burden; nursing; meta-analysis; mental health; health questionaries

## 1. Introduction

Non-communicable diseases (NCDs) have a long duration with usually slow progression and affect all age groups, regions, and countries [1]. They are associated with older age groups, but data show that 15 million of all deaths attributed to NCDs occur between the ages of 30 and 69 years [2]. Fourteen percent (14%) of the global burden of disease can be attributed to mental and neurological disorders. In 2008, the World Health Organization (WHO) developed a Mental Health Gap Action Program (mhGAP) for these mental illnesses [3]. Dementia is not a specific disease but rather a general term for the impaired ability to remember, think, or make decisions that interferes with completing everyday activities. Alzheimer's disease is the most common type of dementia. Though dementia mostly affects older adults, it is not a part of normal aging [4]. At the WHO Executive Board held in the first quarter of 2019, Alzheimer's Disease International (ADI)

presented two important statements on non-communicable diseases (NCDs) and Universal Health Coverage (UHC), with the aim of bringing dementia to the forefront [5].

Dementia is a severely disabling disease for those who suffer from it and is often devastating for caregivers and family members. Currently, 35.6 million people are living with dementia worldwide, and this number is on track to almost double by 2030 and more than triple by 2050 [6]. In 2010, the total estimated cost of dementia was USD 604 billion, but in contrast, in lower–middle income countries, direct social costs were small, with informal care costs dominating [6].

Dementia has an impact on the family, particularly on the caregiver. Most care is provided by families, usually women, and by other non-formal support systems in the community [7]. However, demographic changes in the population could reduce the availability of informal caregivers in the future [6]. Seventy-five percent (75%) of family caregivers state that "Between caregiving and other responsibilities, I am often stressed", and more than 50% state that their health has suffered as a result of their caregiving responsibilities. It is clear from the report that even in high-income countries, most categories of respondents felt that there were not enough services available [8]. There are interventions to reduce caregiver burnout and, increasingly, they are provided through new technological tools. Thirty-six percent (36%) of respondents said they would seek this help on the Internet [8]. Where care is formal, within the health and social care sector, there are different levels of assistance offered to people with dementia that can help to reduce burnout on the caregiver. Within these levels of primary and community care, it is possible to find general practitioners, nurses, and social workers [6]. These professionals fulfill the role of social case management, establishing a strong link with the person with dementia and their family, providing access to the various support resources available [6,9]. Caregivers experience greater burden than non-caregivers. Caregivers are more stressed, depressed, and have lower levels of subjective well-being, physical health, and self-efficacy [10]. Brodaty et al. [11] concluded that the different interventions provided have a significant effect on caregiver health, may reduce their psychological morbidity, and help people with dementia to stay at home for longer. The type of interventions provided to caregivers to improve their health range from educational actions (group and individual) to on-demand interventions with support groups [12]. Support groups are seen to benefit caregivers, so special emphasis needs to be placed on a better understanding of the effectiveness of support groups in the future [13]. Jensen et al. [14] conclude in their systematic review that educational programs have a moderate positive effect on caregiver burden and a minor positive effect on depression.

In the reviewed studies, there are different health questionnaires to adequately assess caregiver burden in patients with dementia, for example, the Zarit scale [13,14]. The internal structure and reliability of these questionnaires is currently being studied for other types of patients, such as those with intellectual disability [15].

The various published reviews present contradictory results and great difficulty in data collection, due to the use of different scales, heterogeneous population groups, presentation of partial results, and different final results (mean, standard deviation, difference of means, *p*-values, etc.). Integrating the different studies in an appropriate way is challenging due to their variability. Several systematic reviews recommend that future research should seek to improve clinicians' ability to prescribe interventions [11] and the need for more high-quality Randomized Control Trials (RCTs) for specific caregiver groups [15]. The main objective of the proposed review was to determine which type of intervention is the most appropriate to improve the health of the primary caregiver in patients with dementia and which factors were important in explaining the heterogeneity. Other objectives were to find out whether the role played by the nurse in multidisciplinary teams is a determining factor in the effectiveness of these interventions and to understand whether the different questionnaires used in the studies measure caregiver health in the same direction.

Based on the above, this paper seeks to identify which interventions can improve the health of the primary caregiver of patients with dementia, helping the multidisciplinary

teams that apply the health intervention. It also aims to note which moderating variable influences the intervention to have the best possible outcome. Finally, we intend to show how the use of similar measured scales may have a minor effect on the interpretation of the final result of the research.

## 2. Materials and Methods

This systematic review and meta-analysis were guided through the PRISMA (Preferred Reporting Items for Systematic Reviews and Meta-Analyses) 2020 recommendations [16], and the action protocol was registered in Prospero, PROSPERO 2020 CRD42020170571. It is available at: https://www.crd.york.ac.uk/prospero/display_record.php?ID=CRD42020 170571 (accessed on 9 February 2024).

### 2.1. Literature Search Strategy and Study Selection

The articles were obtained after conducting a systematic search with an initial date of 30 September 2019 to 30 March 2020. The different searches throughout this period were carried out in all databases at the same time.

The meta-analysis has not been updated because in a recent search (30 January 2024), no articles with inclusion criteria in relation to the meta-analysis protocol were detected.

The databases used were Google Scholar, Cochrane Library, Wiley Online, and Web of Science (Derwent Innovations Index, KCI-Korean Journal Database, MEDLINE, Russian Science Citation Index, SciELO Citation Index). The search for gray literature was carried out using the Opensigle search engine, currently OpenGrey, of the European SIGLE project. The main combinations in the search for these articles contained the keywords "dementia", "Alzheimer's", "burden", "intervention", "psychosocial", "randomized controlled trial", and "Zarit scale". Another source of information was the use of articles cited in other studies or previous systematic reviews. There was no restriction on the language of the publication, with articles written in English, Spanish, and German.

The inclusion criteria were as follows: (1) randomized controlled trial-type studies, (2) one or several control groups, (3) pre- and post-intervention measures, (4) the health variable assessed must be caregiver burden, (5) any intervention to improve caregiver burden, (6) psychometric instruments that measure this burden, and (7) family caregivers of patients with some type of chronic non-communicable disease of neurological origin, either vascular dementia with stroke or other types of Alzheimer's-type dementia. The following were excluded: (1) articles that were not RCTs, (2) abstracts or incomplete articles, (3) studies that did not assess burden, and (4) studies using instruments with no psychometric relationship.

The high-sensitivity search was elaborated using descriptors according to the population, intervention, comparison, and outcome (PICO) strategy: populations: "Caregiver" OR "Caregiver, Family" (caregiver of person with dementia); intervention: "Education " OR "Support Groups" OR "Activity, Educational"; comparator: methodology vs. controls (involvement with intervention non-technological); pre- vs. post-intervention (interventions planed with exercise and sport); and outcomes: "Caregiver Burden" OR "Zarit Caregiver Burden Scale" OR "Behavior Rating Scale" [17].

All studies had to meet the selection criteria. Firstly, an author developed the selection by reading the titles and abstracts. Next, the selection was made by reading the title and abstract. Subsequently, the full content of the selected articles was obtained.

A second researcher was responsible for reviewing the appropriateness of the articles included. Discrepancies found between the two researchers were discussed and agreed upon, and the agreement between reviewers reached a kappa value of 0.78 [18]. Trials were not excluded based on the quality appraisal; rather, it informed critical evaluations of the conclusions of the included trials.

## 2.2. Variables of Interest

Caregiver burden was assessed with instruments related to each other. That is, some instruments were modified or improved versions of others, or even relied on each other for the creation of a new one. The instruments used were the *Caregiver Burden Index (CBI)* [19], the *Caregiver Strain Scale* [20], the *Impact of Caregiver Scale* [21], the *Memory and Behavior Problems Checklist (MBP)* [22], *Memory and Behavior Problem Checklist (BI)* [23], and the *Revised Memory* and *Behavior Problem Checklist (RMBPC)* [24].

## 2.3. Moderating Variables

Demographic data included sex, age, length of schooling or economic measures of the caregiver, the type of professional who carried out the intervention, the country where the study was carried out, and the quality of the studies according to three different scales (Jadad scale [25], Delphi list [26], and Forbes scale [27]) (Supplementary Tables S7–S9).

The variables related to the design of the studies were the patient's pathology under study, type of intervention carried out, psychometric instrument assessing caregiver burden, partial results of the instrument used, global results of the instrument used, time of the intervention and follow-up of the patient after the end of the intervention, and the type of control groups created.

The identification of the studies was performed using the author's name, date, and a unique identifier code. The studies were grouped (individual information [28–30], individual therapy [31–44], group therapy [33,45–55], respite care [33,56,57], support group [32,45,58–62], and workshop [20,63,64]) according to the type of intervention [12,65] (Supplementary Tables S1–S6).

## 2.4. Data Extraction

Where a study applied different interventions to the same population, the results of that study were associated with each of the interventions used.

Due to the type of instrument and how the comparison between groups was carried out, the results of an improvement in caregivers' health were expressed in both negative and positive changes. For this reason, all scales were recoded to mean that the more positive the scores, the better the caregiver's health.

The studies had to contain sufficient mean, standard deviation, and sample size or results to allow for the calculation of the effect size used.

## 2.5. Data Analysis and Synthesis

The review was conducted under the assumption of the fixed effects model. The effect size calculated was the Hedges standardized mean difference. The test of homogeneity (Cochran's Q test) and the $I^2$ coefficient were calculated to determine the presence of heterogeneity. Given the importance of heterogeneity in meta-analytic studies and the low power of the test, a significance level of 0.10 was used. In the case of heterogeneity detection, the causes of determination were explored by a meta-partitioning process [66]. This method divides the homogeneity test into a sum of squares between and within groups to determine whether a factor explains heterogeneity. It is based on the ideas proposed by Hedges and Olkin [67], and the regression and classification tree methods proposed by Breiman [68]. For the detection of publication bias, the funnel plot was used together with Egger's test [69].

Version 2 of the Cochrane ROB (risk of bias) tool for randomized trials [70] was used to evaluate the ROB for each study.

The data processing for the quantitative review was performed with the R package 'metafor' [71], MAd package [72], RcmdrPluggin MA package [73], and SAS-JMP version 12 [74].

## 3. Results

The search yielded 1512 records, of which 338 were identified in databases and 1174 in qualitative–quantitative reviews. A total of 32 records articles were eliminated. A review of the titles and abstracts was performed on 275 reports. The full text of 39 articles that met the inclusion criteria was reviewed (Figure 1).

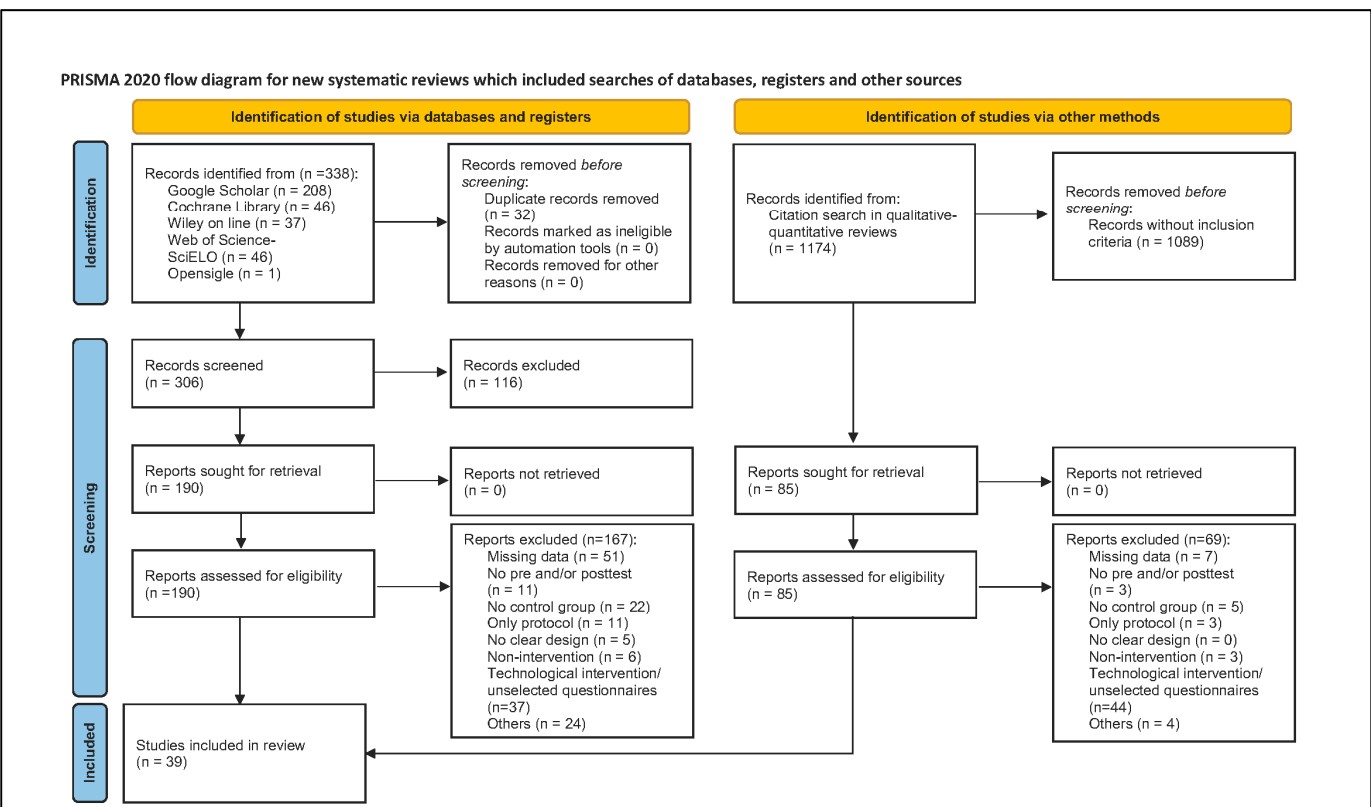

**Figure 1.** PRISMA flow chart [16].

### 3.1. Description of the Studies

The 39 studies had a total sample of 4715 participants (caregivers). The mean age of the caregiver was 59.92 years (SD: 6.87; max: 73.7; min: 40.63), and the majority of the participants were (72%) female. Years of education of the caregiver were described quantitatively in 17 studies, with a mean of 11.54 years (SD: 3.66; max: 15.7; min: 0.68). The mean income of the patient's family was USD 19,060.15 (SD: USD 11,605.48; max: USD 32,132.64; min: USD 250.85) in a total of six studies. In relation to patient pathology, 36% of the studies selected only patients with a diagnosis of Alzheimer's disease. The remaining 64% had mixed pathologies: (1) patients with a diagnosis of Alzheimer's or other dementia, 39.5%; (2) other types of well-defined dementias, 1.5%; and (3) pull of patients with ill-defined dementias, 59%. In 31 articles, the caregiver–patient relationship was spouse in 48% (61% were wives) and children in 35% (84% were women). Other relationships made up the remaining 17%, ranging from sibling to friend. When the caregiver was a sibling, 67% were male. In 77% of cases, the individuals providing the intervention were health professionals [75]; 5% were non-professionals or facilitators. In five studies, the sample size was greater than 80 individuals. The intervention had a mean duration of 11 weeks (SD: 8; max: 52; min: 1). In 17 studies where follow-up was conducted, the mean duration was 17 weeks (SD: 10.57; max: 52; min: 6).

### 3.2. Aggregation of Results

The integration of the results in the different interventions shows that the effect size is small (Figures 2–4), with the presence of heterogeneity in most of the interventions

except for "*Individual information*", where the effect is moderate. The effect size in this intervention is 0.48 (CI95%: 0.18; 0.79; k = 3) with no heterogeneity (Q = 0.78, *p* = 0.676, I2 = 0%). "*Group therapy*", "*Workshop*", and "*Respite care*" interventions show small effect sizes: 0.20 (CI95%: 0.08; 0.31; k = 12), 0.21 (CI95%: 0.01; 0.42; k = 3), and 0.22 (CI95%: 0.05; 0.40; k = 3), respectively. The heterogeneity detected in "*Group therapy*" was explained (Q = 12.06, *p* = 0.796, I2 = 0%) by removing one article [52] with a larger effect size than the rest of the studies (Figure 5). Also, the heterogeneity was significant in the "*Workshop*" intervention (Q = 9.54, *p* = 0.089, I2 = 48%). If we analyze the causes of heterogeneity in this intervention, we find that in studies in which nurses participate, the intervention is more beneficial (0.32 (CI95%: 0.01; 0.54); Q = 2.72, *p* = 0.436, I2 = 0%) compared to studies in which they do not participate (−0.32 (CI95%: −0.81; 0.17); Q = 1.34, *p* = 0.247, I2 = 25%) (Figure 5). Heterogeneity is also found in "*Respite care*" (Q = 17.53, *p* = 0.008, I2 = 66%.). One article [57] presents the results of the different dimensions of the instrument separately. The heterogeneity is explained (Q = 0.0, *p* = 0.98, I2 = 0%) when this article is removed (Figure 5). The effect size in "*Individual therapy*" was also small, at 0.28 (CI95%: 0.15; 0.42; k = 14), with moderate presence of heterogeneity (Q = 63.24, *p* < 0.001, I2 = 68%). The best moderate factor explaining the heterogeneity in this intervention is the country factor. This factor defines two groups of countries (Mediterranean/Latin: Brazil, Spain, Italy, and Turkey, and other: Germany, China, and the USA) with important differences in effect size and low heterogeneity (Figure 5): (1) studies from Mediterranean/Latin countries (0.74 (CI95%: 0.43; 1.04); Q = 0.52, *p* = 0.914, I2 = 0%) and (2) studies from other countries (0.11 (CI95%: −0.04; 0.26); Q = 16.74, *p* = 0.334, I2 = 10.4%). As this factor could be affected by a confounding factor, we analyzed the characteristics of the studies from these two sets of countries and found that the articles published by Mediterranean/Latin countries had a higher number of sessions per week and number of weeks of intervention. In addition, caregivers also had lower mean ages. Finally, in the "*Support Group*", the effect was almost null 0.07 (95%CI: 0; 0.15; k = 6), with significant heterogeneity (Q = 32.32, *p* < 0.001, I2 = 78%). It did explain some of the heterogeneity in this group (Q = 9.28, *p* = 0.159, I2 = 35%) when one article [61] was removed (it had a much higher effect than the rest and uses an instrument (CBI) translated into the local language (Chinese) which is not used by the rest of the articles in this group) (Figure 5). The factor related to the questionnaire used has never been important in explaining the heterogeneity of the results. The rest of the methodological factors did not affect the heterogeneity found in the different interventions. Despite the wide range of years of publication of the studies, no association was detected with the magnitude of the effect on the different interventions. To confirm that there were no changes in the type of design over time, an association analysis of the years with the design variables was carried out. Only the % of dropouts was statistically significant. The correlation between the % of dropouts and years was negative (r = −0.58; *p* < 0.001), so that in the first years, the % of dropouts was higher than in recent years. Given this result, the correlation between the treatment effect and the effect sizes in the different interventions was also measured. No significant association was detected, although in the group therapy intervention, the correlation coefficient was moderate (r = −0.57; *p* = 0.37).

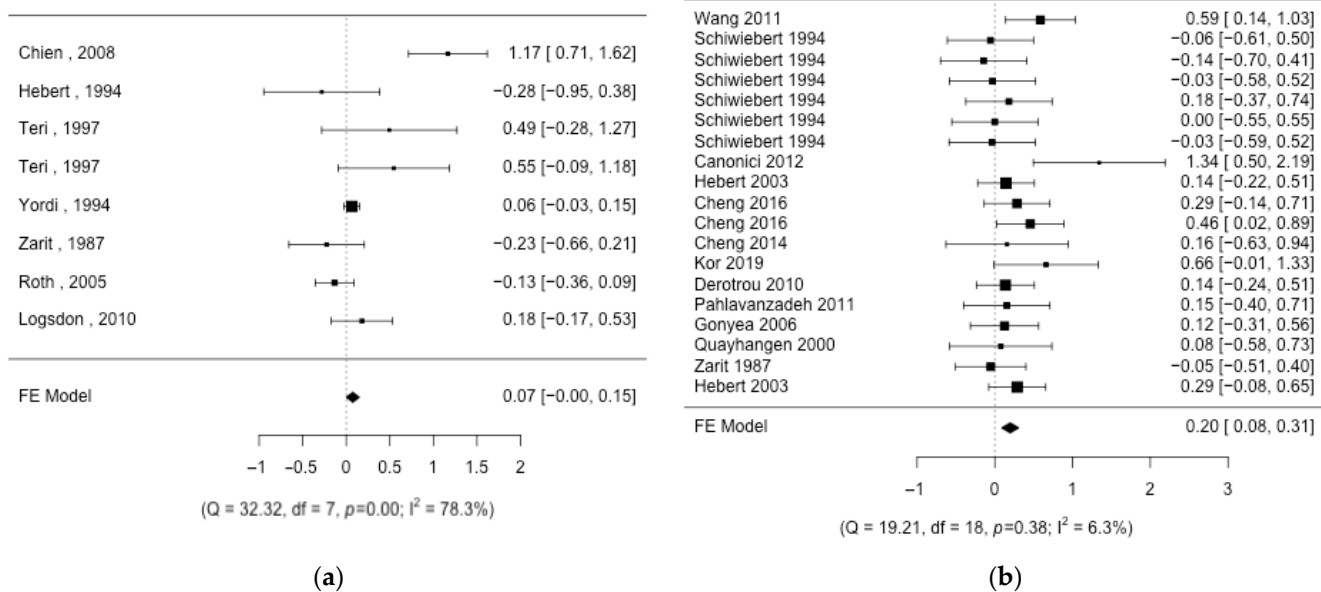

(**a**)

(**b**)

**Figure 2.** Forest plot of the different intervention groups. (**a**) Support group; (**b**) group therapy [32,45–55,58–62].

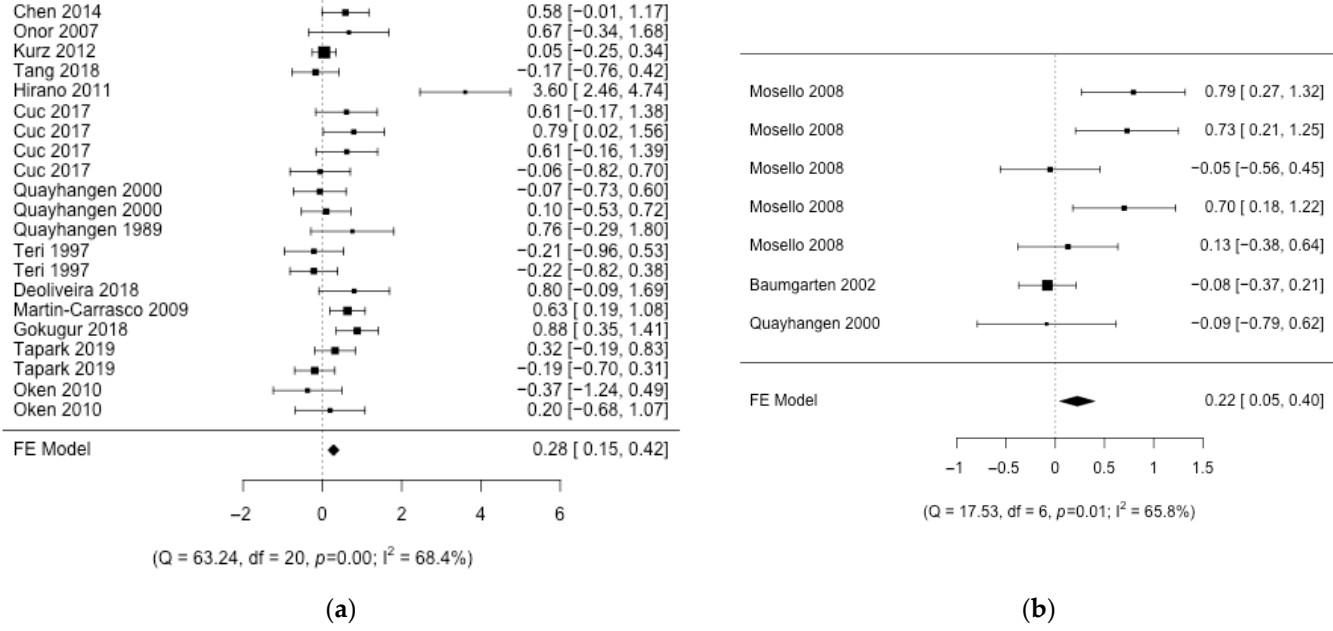

(**a**)

(**b**)

**Figure 3.** Forest plot of the different intervention groups. (**a**) Individual therapy; (**b**) respite care [31–41,43,44,56,57].

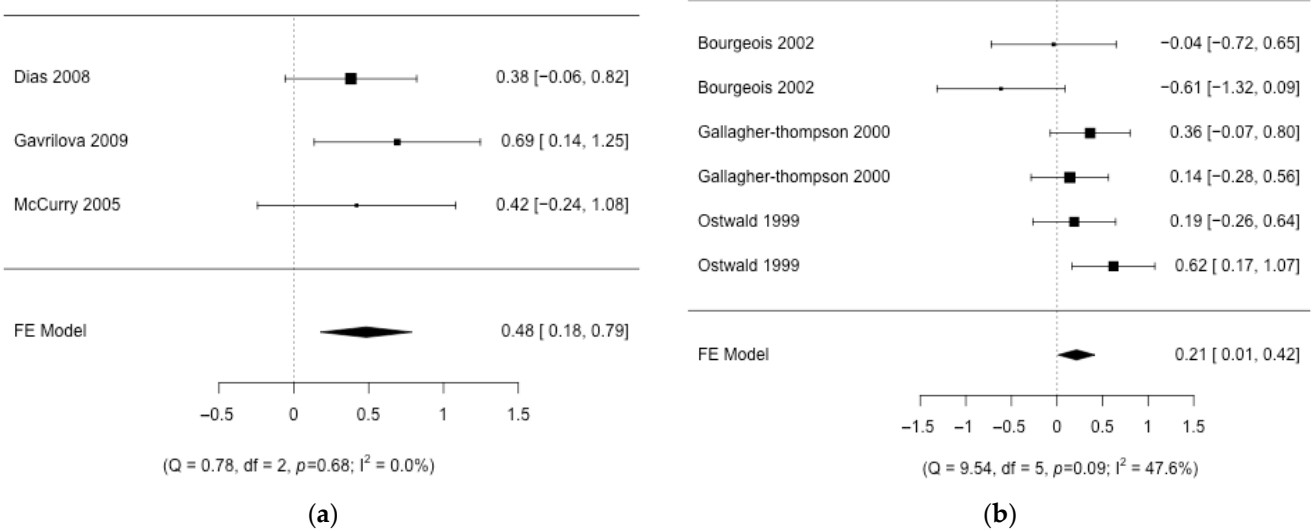

**Figure 4.** Forest plot of the different intervention groups. (**a**) Individual information; (**b**) workshop [20,28–30,63,64].

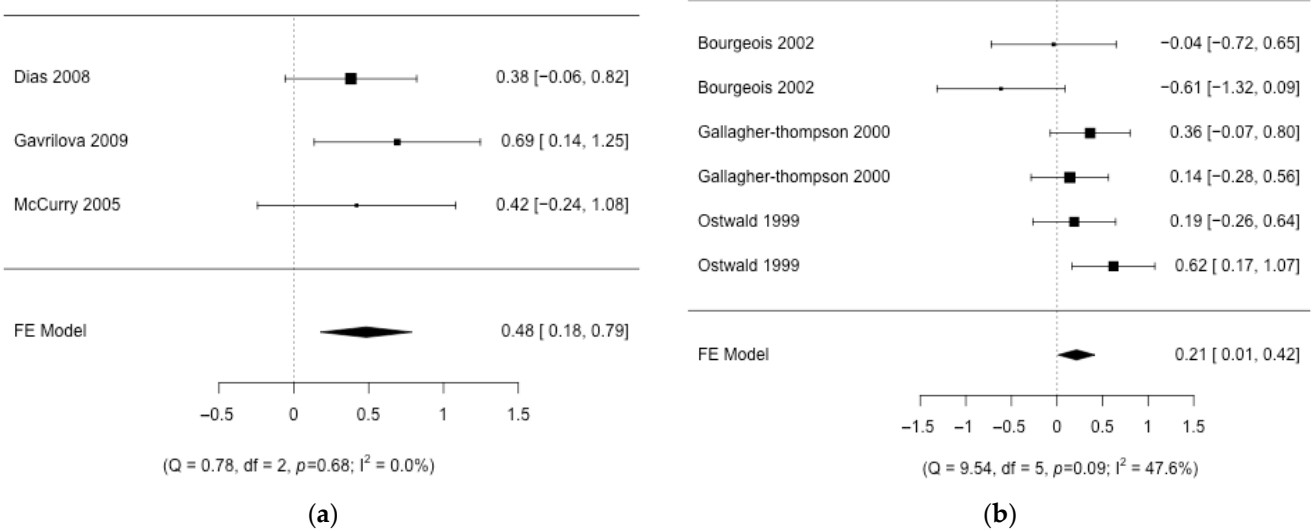

**Figure 5.** Forest plot of the different intervention groups after analyzing the origin of the existing heterogeneity.

### 3.3. Risk of Bias

The analysis of publication bias in the intervention groups with the largest number of studies shows that no publication bias was detected, except in the individual therapy group, where the test result is Egger: $z = 2.8926$, $p = 0.0038$. Considering that this group showed high heterogeneity, these results may be due more to heterogeneity itself than to publication bias.

An analysis of ROB (risk of bias) has also been carried out in the evaluations of each of the trials. See Supplementary Table S10 for a summary (green represents low risk, amber represents unclear risk, and red represents high risk of bias).

## 4. Discussion

The health variable "*burden*" is complex to modify and difficult to measure in the personal and family scope of human relationships [13]. In previous reviews, the effect size was moderate and showed a wide range of values; this range was often confusing as all the interventions studied were combined [13,76–80]. In the present work, the studies are grouped by type of intervention, showing a moderate positive effect in some of them, which does not allow discriminating which intervention is better for improving the health of the main caregiver.

The demographic variables evaluated confirm that the caregiver is usually the wife, followed by the children. In this sense, Dukin et al. [12] commented that the role of the woman in society generates great stress, and women suffer from a greater care burden than men. As a possible confounding factor, women participate more in studies assessing this alteration in caregiver health. The caregivers included in this research are older caregivers whose average age is close to 60. If we look at emerging countries (Brazil, India, Iran, Brazil, Turkey), the average age of the caregiver is lower, around 50 years old. In future studies, it would be interesting to know whether in these countries, where the caregiver population is younger, if the intervention would have more positive effects on the caregiver and improve their overall health status.

The *study quality* variable did not have sufficient weight to explain the *heterogeneity* present in the different groups, but from the initial design, only randomized studies were included because it is a feature that can avoid biases such as overestimating the effect of the intervention. Highly distressed caregivers are more likely to be assigned to the intervention group, and this leads to high dropout rates and missed negative outcomes of the intervention [77]. Our results do not confirm this assertion, although they should be considered with caution because the effect of dropouts was not considered an objective of this meta-analysis. Along with randomization, dropout is an important element in measuring study quality. The dropout rate did not exceed 30%, except for one study [59], but this was not a significant variable in the review. It should be noted that the quality of the studies as a whole should not be evaluated exclusively, since depending on the type of intervention, different types of people are reached with different dropout rates [80].

In our study, the identification of the origin of the control group, i.e., whether it is a control group in essence or whether it is a group on a waiting list or receiving another type of therapy, has been of value. This characteristic can provide insight into the final effect of the intervention [78,80], but in our case it was not conclusive.

The number of sessions per week and duration of the intervention (the effectiveness of the intervention will be better the longer and more frequently it is performed); the region (socio-cultural reasons make the scheduling of the intervention and its effect differ between countries or regions); and the professional delivering the intervention (the presence of nurses suggests an improvement in the effect of the intervention in workshop groups) have proved to be important moderating variables. This last variable has not been included in previous studies as a moderating variable [65], and professionals are essential for the intervention to run smoothly. In some studies, the number of hours or training sessions that professionals or facilitators must receive to deliver the intervention to the caregiver is described [29].

If the outcome of the different interventions is evaluated, the support group has a low, almost null effect on burden, as in other studies [13,78]. The presence of heterogeneity is important and does not change the effect of the intervention after it has been explored and explained. It seems that the support group strategy works because the caregiver leaves the session armed with practical skills [13]. Group therapy, also called family therapy in previous studies, showed a low intervention effect [72], but it is a therapy that brings the clinician and the family into interaction [12]. Individual therapy showed a moderate intervention effect with a strong presence of heterogeneity. Two well-differentiated groups defined as "Mediterranean/Latin" countries responded better compared to the group of "other" countries. In previous reviews, it has been argued that ethnicity and culture are

important factors in whether one intervention or another is effective [81]. Also, the duration of therapy and the age of caregivers in individual interventions can modify the outcome, as seen earlier in this review. The direct intervention of this type of therapy is key to reducing the levels of depression and burden [78], but it should not be forgotten that group therapy can compete with this efficacy due to the relationship and bonds created between its members [80]. Pinquart et al. [77] suggest that active participation is necessary and is linked to the caregiver's attitude. If their attitude is depressive, family overload is added, meaning that participation will not be active. In short, the type of intervention and the caregiver's attitude are fundamental to the effectiveness of any intervention. The individual caregiver information group (which included only three studies) was more effective than the individual therapy group. Interestingly, only transmitting the information without practicing what has been learned improves the caregiver's symptomatology, potentially questioning the effectiveness of complex interventions compared to simple interventions and a possible greater degree of adherence by the caregiver to the latter. When the intervention is oriented towards the respite care, the heterogeneity is high (day center, admission by days, etc.), as in previous studies [78], with a moderate effect on caregiver burden. A factor evaluated in previous studies is the duration of this last intervention, which in our case has been within this range [12]. It should be noted that this type of intervention may have a selection bias due to the fact that in some cultures, there is a tendency not to take the family member to a residential facility [82]. Shame is mainly to blame for the fact that this type of intervention is not very dominant or accepted by caregivers [12] and that few studies are potentially suitable.

A major factor is dose-dependency [65], i.e., the effectiveness of the therapy in relation to the number of days or weeks the intervention lasts [45,78,83–86]. This dose is estimated in the Family Survival Project, where it is stated that the right amount for optimal effectiveness of the intervention is eight hours per week [86]. In this meta-analysis, it has been observed that the higher the number of sessions, the better the response in countries with a "Mediterranean/Latin" culture in the individual therapy group.

A very interesting element registered in this study was knowing how the different integrations of the articles chosen for the meta-analysis are presented, and how this variability can be a factor that influences the heterogeneity of the results. In some studies, the authors seem to mix different instruments [9], but all the health questionnaires/instruments used are validated questionnaires, even in the patient's own native language [53]. Moreover, the type of instrument used in the studies has not been relevant in explaining the high heterogeneity detected in the integration of effect sizes. Other authors explain which data they have extracted according to the symptoms that are assessed [10,77,80]. What is less clear is the integration procedure, if any, when a study uses two similar instruments or delivers partial data from the same instrument. In our study, different instruments have been used, but with similar characteristics in a conscious way. Searching for articles with the same instrument would limit the search. Using studies with very different instruments can be detrimental when integrating results and can be a handicap when analyzing the dependence of the results.

*Limitations*

A first limitation is the difficulty of finding studies of sufficient quality in relation to the blinding of the intervention given the difficulty of carrying out this procedure in this type of study [87,88].

The underestimation of the value of caregiver burden shown by the caregiver is another limitation in these studies [10,78]. In addition, there is a lack of knowledge about the sensitivity of instruments that measure caregiver burden in relation to the patient's disease state. There are studies where authors warn that if an instrument is not sensitive enough to detect changes in certain populations, such as caregivers, the effect of an intervention may be underestimated and, therefore, interventions that may appear beneficial to participants are not considered cost-effective and are not funded [89–91].

The results of the studies are also affected by the lack of information about the patient–caregiver relationship, as well as the patient's social context. Many studies do not even reflect the monthly remuneration received by the assessed family. In the present study, the relationship between the caregiver and the patient has been specified according to the degree of consanguinity but focused on the caregiver without differentiating the stage of the cared patient's illness, a factor that may influence the degree of stress of the caregiver.

Although there are currently quite a few articles that focus on interventions that use technological tools, they have not been considered in this meta-analysis.

For future studies and reviews, it is necessary to further analyze the economic level quantitatively [29,61], the size of the intervention group [13], the underestimation of burden [10], and the time of the disease when the study is performed in order to further investigate the real effect on the level of burden.

## 5. Conclusions

The effect size is low, with the presence of high heterogeneity for all interventions except for the "individual information" intervention, where the effect is moderate, and the heterogeneity is not significant. In addition, it would be necessary to study what magnitude of effect size would have a relevant significance on caregiver burden. This meta-analysis has also shown that the length of time the intervention lasts, the age of the caregiver, what community services are available in particular geographic regions, and the stage of the patient's illness are factors that should be clearly shown in the primary studies to be analyzed in future reviews since several of the articles included in this meta-analysis did not present such information. The type of questionnaire used to measure caregiver health was not a determining factor in explaining the differences between effect sizes.

The role of the nurse is essential when interventions are carried out to reduce the stress of the main caregiver of people with dementia. The nurse participates in many types of interventions and multidisciplinary groups. This study has detected that the presence of nurses in these multidisciplinary groups improves the effect of the intervention on caregivers when it is carried out in the form of workshops, although this does not imply that the role of nurses is not important in the rest of the interventions because the results were not statistically significant. The closeness and continuity of care that nurses provide is a historically recognized fact that may be related to this positive result. On the other hand, the presence of nurses is identified as a moderating variable that will help in future studies to detect other factors that influence the improvement in the results of the interventions.

**Supplementary Materials:** The following supporting information can be downloaded at: https://www.mdpi.com/article/10.3390/nursrep14020071/s1, Table S1. Respite care studies; Table S2. Support group studies; Table S3. Individual information studies; Table S4. Workshop studies; Table S5. Group therapy studies; Table S6. Individual therapy studies; Table S7. Jadad scale quality studies; Table S8. Delphi list quality studies; Table S9. Forbes scale quality studies; Table S10. Risk of bias (ROB) 2.0; Table S11. The PRISMA Checklist; Figure S1. Heterogeneity evolution when excluding articles; Figure S2. Heterogeneity evolution when excluding articles.

**Author Contributions:** Conceptualization, F.J.R.-A. and J.M.-V.; methodology, F.J.R.-A. and J.M.-V.; software, F.J.R.-A. and J.M.-V.; validation, F.J.R.-A., R.J.-V., J.L.S.-G. and J.M.-V.; formal analysis, F.J.R.-A. and J.M.-V.; investigation, F.J.R.-A. and J.M.-V.; resources, F.J.R.-A. and J.M.-V.; data curation, F.J.R.-A. and J.M.-V.; writing—original draft preparation, F.J.R.-A., R.J.-V., J.L.S.-G. and J.M.-V.; writing—review and editing, F.J.R.-A., R.J.-V., J.L.S.-G. and J.M.-V.; visualization, F.J.R.-A., R.J.-V., J.L.S.-G. and J.M.-V.; supervision, R.J.-V., J.L.S.-G. and J.M.-V.; project administration, F.J.R.-A. and J.M.-V.; funding acquisition, No funding required for this study. All authors have read and agreed to the published version of the manuscript.

**Funding:** This research received no external funding.

**Institutional Review Board Statement:** Not applicable.

**Informed Consent Statement:** Not applicable.

**Data Availability Statement:** The datasets generated during and/or analyzed during the current study are available from the corresponding author on reasonable request.

**Public Involvement Statement:** No public involvement in any aspect of this research.

**Guidelines and Standards Statement:** This manuscript was drafted against the PRISMA (Preferred Reporting Items for Systematic Reviews and Meta-Analyses) 2020 recommendations.

**Conflicts of Interest:** The authors declare no conflicts of interest the manuscript.

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
