# Peer review of "Interventions Effective in Decreasing Burden in Caregivers of Persons with Dementia: A Meta-Analysis"

_nursrep, doi:10.3390/nursrep14020071_

Round 1
Reviewer 1 Report
Comments and Suggestions for Authors
This paper describes the significance of dementia and associated caregiver burden. It identifies ranges of intervention groups for caregivers of dementia and assesses for impact on caregiver health. This study holds considerable promise and contribution to the field and, addressing some comments below may help bolster the paper:
1) "Respire care" was one of the intervention groupings. Do you mean respite care?
2) About 64% had mixed patient pathology, which may in itself contribute to inherent variability of caregiver burden. For instance, frontotemporal dementia and early-onset dementia occurs at a younger age compared to Alzheimer's disease. As such, caregiver burden may differ across different types of dementia. Have you considered including similar pathology characteristics (e.g., age of onset) when narrowing down the articles included in the analysis?
3) When country of origin of these studies were classified by Mediterranean/Latin vs Continental, it may be important to consider the cultural differences between Germany, China, and USA. What is the rationale behind comparing these studies by Latin vs Continental?
4) Regarding the "workshop" intervention group, along with the inclusion/exclusion of nurses, have you considered other paid caregivers (e.g., home health aides) in your assessment?
Author Response
REVIEWER 1
Thank you for the time invested in the review process and for providing thoughtful feedback on our manuscript. Below, we have addressed each of your comments point-by- point and revised the manuscript accordingly. All the amendments in the revised manuscript are highlighted. After revising the manuscript according to your suggestions, we feel that its quality has been substantially improved. We hope you consider how we addressed the comments positively, and we look forward to hearing your decision.
Comment 1: "Respire care" was one of the intervention groupings. Do you mean respite care?
Thank you for your suggestions. Indeed, respite care provides short-term relief for primary caregivers, giving them time to rest, travel or spend time with other family and friends.
We changed the concept.
Comment 2: About 64% had mixed patient pathology, which may in itself contribute to inherent variability of caregiver burden. For instance, frontotemporal dementia and early-onset dementia occurs at a younger age compared to Alzheimer's disease. As such, caregiver burden may differ across different types of dementia. Have you considered including similar pathology characteristics (e.g., age of onset) when narrowing down the articles included in the analysis?
Thank you for your insightful observations. We agree with his statement, but this criterion was not considered for the selection of articles because of this information is not clearly indicated in the articles. But we realized that it may be important to control the heterogeneity of the results and therefore we included it as a piece of data that should be provided in future research (line 404-414, conclusions section)
“This meta-analysis has also shown that the length of time the intervention lasts, the age of the caregiver, what community services are available in particular geographic regions, and the stage of the patient's illness are factors that should be clearly shown in the primary studies to be analyzed in future reviews since several of the articles included in this meta-analysis did not present such information.”
Comment 3: When country of origin of these studies were classified by Mediterranean/Latin vs Continental, it may be important to consider the cultural differences between Germany, China, and USA. What is the rationale behind comparing these studies by Latin vs Continental?
The country variable presented many categories and therefore could present problems of degrees of freedom in the analysis of the explanation of heterogeneity, so it was decided to group countries. In the case of Mediterranean countries, the grouping was clear; however, for the other set of countries it was not easy to include a name that included common characteristics of these countries. It was decided to define them as continentals as inland countries although the USA is a continent and China is a large country with many particularities. The reviewer is right that this denomination is not the most accurate. Perhaps it would be better to indicate a more undefined category of "Other".
We changed the concept.
Comment 4: Regarding the "workshop" intervention group, alonl vg with the inclusion/exclusion of nurses, have you considered other paid caregivers (e.g., home health aides) in your assessment?
Thank you for pointing out this question. Effectively, other occupational groups were tested to explain the heterogeneity (Social Worker, Therapist, Psychologist, Psychiatrist and Occupational Therapist). As the article indicates, only this paid occupational group explained the presence of heterogeneity.

Reviewer 2 Report
Comments and Suggestions for Authors
This review identifies important factors to inform the design of future research into decreasing caregiver burden by recognising the importance of context (e.g. geographic region, stage of dementia). Identifying the moderating variables is also useful for understanding the differences that can influence how to support caregivers in different contexts. The review methods appear thorough and transparent.
Clarification of whether the meta-analysis been updated would be helpful, as it appears to have been several years since its completion.
The meaning of ‘measure in the same direction’ with regard to questionnaires would benefit from being clearly explained.
Introduction / abstract: the focus on chronic non-communicable diseases would benefit from being situated within the broader context of dementia as defined by the Centers for Disease Control and Prevention. The distinction between diseases that can lead to dementia, such as Alzheimer's, and dementia as an umbrella term for a set of symptoms that can have variety of origins (including brain injury) is not clearly articulated.
The term ‘interesting’ is used at several points as a justification for enquiry (lines: 15, 272, 317, 333). These points would benefit from unpacking to identify how enquiry would address gaps in research / the needs of people affected by dementia, i.e. why they would be interesting and to whom.
Justification for the study aims/objectives is not clearly stated in the introduction - why is the knowledge sought needed and what could the consequences be? The introduction would benefit from a summary of the need for, and potential impact of the research, e.g. the potential contribution to the lives of caregivers and the field of research findings.
The …’great difficulty in data collection’ (line 77-8) would benefit from elaboration as it is unclear why this is mentioned or how it relates to the preceding and subsequent narrative.
2.1 Literature search strategy and study selection: the latest date for literature inclusion is 2020, now four years ago. Perhaps the authors could consider offering the reader an explanation as to why this is, e.g., no new relevant literature has been published.
3 Results (lines 174-177) the decision points of the PRISMA (figure 1) and the PRISMA statement itself would benefit from further explanation/clarity. For example, 120 articles are reduced to 39 at the ‘eligibility’ stage, with no reason for exclusion given. Why is there an arrow connecting ‘records duplicates removed’ to ‘articles excluded’ at the screening stage? How does the large number of articles from the ‘identification’ stage become 391 after the removal of only 32 duplicates? I would not be able to replicate the screening process based on the given information.
4. Discussion: it is not clear whether the points at which the different forms of intervention are implemented is considered - e.g. respite care is often when people are at crisis point and offers a very different form of support to peer support groups. It could be that respite prevents complete crisis as much as reducing existing burden. Have these aspects been considered?
Line 229 – the sentence “The factor related to…” would benefit from unpacking, as the meaning is not clear. This (I assume) relates to the measuring health in the same ‘direction’ mentioned previously the would benefit from explanation.
Line 269 – “…caregivers who are close to 60 years of age” is somewhat vague – would it be possible to be more specific, e.g. whose average age is close to 60?
Lines 269-273 – the discussion of caregivers. Are these caregivers generally or for people living with dementia? It would be helpful if this could be clarified and any implications, e.g. the latter may be older than carers generally as the incidence of dementia increases with age. It would also be helpful to know explicitly if any ‘emerging countries’ are included in the analysis or whether this is identified as an ‘interesting’ area of future research because ‘emerging countries’ are not included.
Line 278 – discussion of drop out rate. Could a high drop out rate not also be an indication of intervention quality / effect - has this been considered?
Line 315 – the conclusion is drawn that the attitude of the caregiver is a fundamental factor to the effect of any intervention. I cannot find any supporting evidence stated for this conclusion to be drawn.
Comments on the Quality of English Language
The term ‘Respire care’ is used throughout and in the Supplementary Materials – I believe this an error and meant to be ‘Respite care’
The first-person plural ‘we’ is used intermittently (e.g. lines 69 and 221) to refer to the researchers and is not consistently. The authors might consider using third person consistently throughout as this may be frowned upon by some academic readers.
Line 19 – “The search…” the meaning of this sentence is not clear
Line 40 - the acronym WHO needs to be defined when it first appears in line 38, rather than in line 40.
Line 53 – the quote would benefit from either: 1. explaining how 75% of family caregivers can all state the same thing (answer to questionnaire?), or 2. paraphrasing the quote.
Line 59 - burnout ‘on’ the caregiver – is this intended to be ‘of’?
Line 57 Examples of formal health and social care professionals are given (“Examples include…”). The preceding sentence leads the reader to expect examples of levels of assistance, which do not appear until the subsequent sentence. These relationship between these three sentences would benefit from some attention for clarity.
Line 63 - the term ‘fare worse’ reads as somewhat informal. Perhaps something more along the lines of ‘experience greater burden’?
Line 64 – “It is also concluded…” Who is concluding or where the conclusion is presented is not articulated – referenced but not articulated.
Line 69 – “It is observed that support groups effect caregivers…” referenced but the source of the observation is not articulated, nor if the type of effect – e.g. positive impact on burden
Line 71 – “In a later review…” there is no review specified within the text prior to this statement to refer back to.
Line 121-2 – two sentences beginning ‘firstly’ / ‘first’. This looks like an error and the meaning is unclear.
Line 290-91 – the meaning of the text in parentheses “(effectiveness will…” is not clear
Line 325 – “Embarrassment is mainly to blame for…” more formal terminology could be more fitting here
Line 347-348 – I cannot make sense of this sentence.
Supplementary tables: Table S3 Group therapy studies – mislabelled? Should this be S5?
Author Response
REVIEWER 2
Thank you for the time invested in the review process and for providing thoughtful feedback on our manuscript. Below, we have addressed each of your comments point-by- point and revised the manuscript accordingly. All the amendments in the revised manuscript are highlighted. After revising the manuscript according to your suggestions, we feel that its quality has been substantially improved. We hope you consider how we addressed the comments positively, and we look forward to hearing your decision.
Thank you for your suggestions. Thank you for your insightful observations.
- Comments and Suggestions for Authors (Reviewer).
Comment 1: Clarification of whether the meta-analysis been updated would be helpful, as it appears to have been several years since its completion.
Thank you for your insightful observations. The meta-analysis has not been updated because in a recent search (start 2024) no articles with inclusion criteria in relation to the meta-analysis protocol have been detected. We include this disclaimer.
Lines 114-116:
The meta-analysis has not been updated because in a recent search (January 30, 2024) no articles with inclusion criteria in relation to the meta-analysis protocol have been detected.
Comment 2: The meaning of ‘measure in the same direction’ with regard to questionnaires would benefit from being clearly explained.
Due to the type of questionnaire, its construction or how the comparisons were carried out, negative or positive values could indicate the same thing, better health in the caregivers. To avoid inconsistencies in the integration between positive and negative values, we transformed all of them into positive values.
Lines 176-179:
Due to the type of instrument and how the comparison between groups was carried out, the results of improvement in caregivers' health were expressed in both negative and positive changes. For this reason, all scales were recoded to mean that positive scores, the better the caregiver's health.
Comment 3: Introduction / abstract: the focus on chronic non-communicable diseases would benefit from being situated within the broader context of dementia as defined by the Centers for Disease Control and Prevention. The distinction between diseases that can lead to dementia, such as Alzheimer's, and dementia as an umbrella term for a set of symptoms that can have variety of origins (including brain injury) is not clearly articulated.
We value the suggested change.
Line 41-44:
Dementia is not a specific disease but is rather a general term for the impaired ability to remember, think, or make decisions that interferes with doing everyday activities. Alzheimer’s disease is the most common type of dementia. Though dementia mostly affects older adults, it is not a part of normal aging [4]. At the WHO Executive Board held in the first quarter of 2019, Alzheimer's Disease International (ADI) presented two important statements on Non-Communicable Diseases (NCDs) and Universal Health Coverage (UHC), with the aim of bringing dementia to the forefront [5].
Comment 4: The term ‘interesting’ is used at several points as a justification for enquiry (lines: 15, 272, 317, 333). These points would benefit from unpacking to identify how enquiry would address gaps in research / the needs of people affected by dementia, i.e. why they would be interesting and to whom.
Thank you for your suggestions.
Line 14-15:
Altering the abstract would alter the maximum total allowed by the journal. We review the phrase.
It is interesting that the different questionnaires measured in the same direction, given that different health questionnaires are used to measure caregiver burden.
Lines 298-301:
Explained below (section/comment 14 in the reviewer's review) in relation to the age of caregiver and emerging countries.
In future studies, it would be interesting to know whether in these countries, where the caregiver population is younger, if the intervention would have more positive effects on the caregiver and improve their overall health status.
Lines 348-352:
Interestingly, only transmitting the information without practicing what has been learned improves the caregiver's symptomatology, potentially questioning the effectiveness of complex interventions compared to simple interventions and a possible greater degree of adherence by the caregiver to the latter.
Lines 367-368:
A very interesting element registered in this study was knowing how the different integrations of the articles chosen for the meta-analysis are presented, and how this variability can be a factor that influences the heterogeneity of the results.
Comment 5: Justification for the study aims/objectives is not clearly stated in the introduction - why is the knowledge sought needed and what could the consequences be? The introduction would benefit from a summary of the need for, and potential impact of the research, e.g. the potential contribution to the lives of caregivers and the field of research findings.
Thank you for your comment. We incorporate elements that improve the content.
Lines 98-103
Based on the above, this paper seeks to identify which interventions can improve the health of the primary caregiver of patients with dementia, helping the multidisciplinary teams that apply the health intervention. It also seeks to warn which moderating variable influences the intervention to have the best possible outcome. Finally, it is intended to show how the use of similar measured scales may have a minor effect on the interpretation of the final result of the research.
Comment 6: The …’great difficulty in data collection’ (line 77-8) would benefit from elaboration as it is unclear why this is mentioned or how it relates to the preceding and subsequent narrative.
Thank you for your insightful observations.
Lines 86-88
The different published reviews present contradictory results and great difficulty in data collection, due to the use of different scales, heterogeneous population groups, presentation of partial results, different final results (mean, standard deviation, difference of means, p-values ...) .
Comment 7: 2.1 Literature search strategy and study selection: the latest date for literature inclusion is 2020, now four years ago. Perhaps the authors could consider offering the reader an explanation as to why this is, e.g., no new relevant literature has been published.
Thank you for your comment. In a systematic review it is difficult to have it updated at the time of publication. We set the dates for article review in the protocol, but it takes time to obtain the information from the articles, process it and prepare the article. However, a review was carried out before submitting the article to see if there were new relevant studies that could affect our results. That review was done in January 2024. It is specified in section/comment 1 in the reviewer's review.
Lines 114-116:
The meta-analysis has not been updated because in a recent search (January 30, 2024) no articles with inclusion criteria in relation to the meta-analysis protocol have been detected
Comment 8: 3 Results (lines 174-177) the decision points of the PRISMA (figure 1) and the PRISMA statement itself would benefit from further explanation/clarity. For example, 120 articles are reduced to 39 at the ‘eligibility’ stage, with no reason for exclusion given.
Thank you for your comment.
The number of articles was reduced from 120 to 39 due to the selection of those articles that presented scales related to caregiver health and those that considered technological therapies were also discarded.
We have updated and adapted the flowchart to the new standards indicated by the Assistant Editor. We hope that the changes help the reader of the content.
Page, M. J.; McKenzie, J. E.; Bossuyt, P. M.; Boutron, I.; Hoffmann, T. C.; Mulrow, C. D.; Shamseer, L.; Tetzlaff, J. M.; Akl, E. A.; Brennan, S. E.; Chou, R.; Glanville, J.; Grimshaw, J. M.; Hróbjartsson, A.; Lalu, M. M.; Li, T.; Loder, E. W.; Mayo-Wilson, E.; McDonald, S.; McGuinness, L. A.; Stewart, L. A.; Thomas, J.; Tricco, A. C.; Welch, V. A.; Whiting, P.; Moher, D. The PRISMA 2020 Statement: An Updated Guideline for Reporting Systematic Reviews. BMJ 2021, n71. https://doi.org/10.1136/bmj.n71
http://prisma-statement.org/
http://www.prisma-statement.org/PRISMAStatement/FlowDiagram
Comment 9: Why is there an arrow connecting ‘records duplicates removed’ to ‘articles excluded’ at the screening stage?
We have updated and adapted the flowchart to the new standards indicated by the Assistant Editor. We hope that the changes help the reader of the content.
Comment 10: How does the large number of articles from the ‘identification’ stage become 391 after the removal of only 32 duplicates? I would not be able to replicate the screening process based on the given information.
We have updated and adapted the flowchart to the new standards indicated by the Assistant Editor. We hope that the changes help the reader of the content.
Comment 11: 4. Discussion: it is not clear whether the points at which the different forms of intervention are implemented is considered - e.g. respite care is often when people are at crisis point and offers a very different form of support to peer support groups. It could be that respite prevents complete crisis as much as reducing existing burden. Have these aspects been considered?
Thank you for your insightful observations.
This aspect has not been considered as there are only three articles with this inclusion characteristic, and the degree of variability in the characteristics of the participants. Reviewing the question you raise, only the article presents the characteristics about which you ask the question.
Baumgarten, M.; Lebel, P.; Laprise, Hé.; Leclerc, C.; Quinn, C. Adult Day Care for the Frail Elderly. J. Aging Health 2002, 14 (2), 237–259. https://doi.org/10.1177/089826430201400204.
“Clients with urgent needs were not included in the study because it was ethically unacceptable for them to spend 3 months on the waiting list in the event that they were randomized to the control group. Clients were considered to have urgent needs if they were receiving 20 hours or more per week of home care or if their informal caregiver was experiencing an episode of acute ill health at the time of referral.”
Comment 12: Line 229 – the sentence “The factor related to…” would benefit from unpacking, as the meaning is not clear. This (I assume) relates to the measuring health in the same ‘direction’ mentioned previously the would benefit from explanation.
Thank you for your suggestions. We rephrase the phrase, as it creates confusion. This phrase is directed to the instrument/questionnaire used no.
Lines 257:
The factor related to the questionnaire used has never been important to explain the heterogeneity of the results.
Comment 13: Line 269 – “…caregivers who are close to 60 years of age” is somewhat vague – would it be possible to be more specific, e.g. whose average age is close to 60?
Thank you for your correction. We have changed the text.
Lines 296-297:
The caregivers included in this research are older caregivers whose average age is close to 60.
Comment 14: Lines 269-273 – the discussion of caregivers. Are these caregivers generally or for people living with dementia? It would be helpful if this could be clarified and any implications, e.g. the latter may be older than carers generally as the incidence of dementia increases with age. It would also be helpful to know explicitly if any ‘emerging countries’ are included in the analysis or whether this is identified as an ‘interesting’ area of future research because ‘emerging countries’ are not included.
Lines 296-301
The caregivers included in this research are older caregivers whose average age is close to 60. If we look at emerging countries (Brazil, India, Iran, Brazil, Turkey), the average age of the caregiver is lower, around 50 years old. In future studies, it would be interesting to know whether in these countries, where the caregiver population is younger, if the intervention would have more positive effects on the caregiver and improve their overall health status.
Comment 15:- Line 278 – discussion of drop out rate. Could a high drop out rate not also be an indication of intervention quality / effect - has this been considered?
Thank for your insightful observations. The dropout element is a controversial factor. It has been included within the quality criteria of the articles and as a moderate variable without being a significant variable in the heterogeneity of the meta-analyses carried out. The limitation with missing data is that the articles provide little information, when they do provide it. It is therefore difficult to analyze the real effect of missing data on the final results. It would be interesting to do a study with this objective in a systematic review.
Comment 16: Line 315 – the conclusion is drawn that the attitude of the caregiver is a fundamental factor to the effect of any intervention. I cannot find any supporting evidence stated for this conclusion to be drawn.
In the article quoted above, it is stated that active participation is necessary, which is linked to the attitude of the caregiver. The attitude of the participant during the intervention is important and therefore active participation is important. Pinquart et al. qualify that active participation on the part of the participant improves the positive outcome obtained and in a particular type of intervention. We adapt the text in a more precise way.
Pinquart, M.; Sörensen, S. Helping Caregivers of Persons with Dementia: Which Interventions Work and How Large Are Their Effects? Int. Psychogeriatrics 2006, 18 (4), 577–595. https://doi.org/10.1017/S1041610206003462.
“Discussion
Caregiver interventions have positive immediate effects on dementia caregivers’ burden, depression, SWB, ability/knowledge and CR symptoms, but only structured multicomponent analyses decrease the risk for institutionalization. However, the effect-sizes are usually small and fewer significant effects appear at follow-up. Psychoeducational interventions have the broadest effects, but only if they call for active participation. Effects of other interventions are mostly domain specific. We found moderator effects of study characteristics and sample characteristics. Our discussion was organized according to the research questions.”
Lines 343-345
Pinquart et al. [77] suggest that active participation is necessary and is linked to the caregiver's attitude. If their attitude is depressive, family overload is added, participation will not be active. In short, the type of intervention and the caregiver's attitude are fundamental to the effectiveness of any intervention.
- Comments on the Quality of English Language.
Comment 17: The term ‘Respire care’ is used throughout and in the Supplementary Materials – I believe this an error and meant to be ‘Respite care’
Thank you for your comment. Corrected in manuscript and supplementary material.
Comment 18: The first-person plural ‘we’ is used intermittently (e.g. lines 69 and 221) to refer to the researchers and is not consistently. The authors might consider using third person consistently throughout as this may be frowned upon by some academic readers.
Thank you for your correction. We have changed the text.
Lines 76-77
Support groups are seen to benefit caregivers and, so special emphasis needs to be placed on a better understanding of the effectiveness of support groups in the future.
Lines 249
As this factor could be affected by a confounding factor, the characteristics of the studies from these two sets of countries are analyzed and found that the articles published by Mediterranean/Latin countries had a higher number of sessions per week and number of weeks of intervention.
Comment 19: Line 19 – “The search…” the meaning of this sentence is not clear
Thank you for your correction. We have changed the text.
Lines 19-20
The systematic search of publish and gray literature was carried out without restriction of the language used in the studies.
Comment 20: Line 40 - the acronym WHO needs to be defined when it first appears in line 38, rather than in line 40.
Thank you for your correction. We have changed the text.
Lines 39,44
In 2008, the World Health Organization (WHO) developed a Mental Health Gap Action Program (mhGAP) for these mental illnesses [3]. … At the WHO Executive Board held in the first quarter of 2019, Alzheimer's Disease International (ADI) presented two important statements on Non-Communicable Diseases (NCDs) and Universal Health Coverage (UHC), with the aim of bringing dementia to the forefront [5].
Comment 21: Line 53 – the quote would benefit from either: 1. explaining how 75% of family caregivers can all state the same thing (answer to questionnaire?), or 2. paraphrasing the quote.
Thank you for your correction. We have changed the text.
Lines 60-61
Seventy-five percent (75%) of family caregivers state that "Between caregiving and other responsibilities, I am often stressed" and more than 50% state that their health suffered as a result of their caregiving responsibilities. It is clear from the report that even in high-income countries, most categories of respondents felt that there were not enough services available [8].
Comment 22: Line 59 - burnout ‘on’ the caregiver – is this intended to be ‘of’?
Thank you for your correction. We have changed the text.
Lines 71-74
That is the intention. Modification included in section/comment 24 in the reviewer's revision also.
Brodaty et al. [11] concluded that the different interventions provided have a significant effect on the caregiver health, may reduce their psychological morbidity and help people with dementia to stay at home longer.
Comment 23: Line 57 Examples of formal health and social care professionals are given (“Examples include…”). The preceding sentence leads the reader to expect examples of levels of assistance, which do not appear until the subsequent sentence. These relationship between these three sentences would benefit from some attention for clarity.
In this case, we are referring to levels of support care such as primary care and community care. Within this attention, the professionals who participate are those expressed.
Thank you for your comment. We have changed the text.
Lines 66-67
Where care is formal, within the health and social care sector, there are different levels of assistance offered to people with dementia that can help to reduce burnout on the caregiver. Within these levels of primary and community care it is possible to find general practitioners, nurses, and social workers [6].
Comment 24: Line 63 - the term ‘fare worse’ reads as somewhat informal. Perhaps something more along the lines of ‘experience greater burden’?
Thank you for your correction:
Lines 70
Caregivers experience greater burden than non-caregivers.
Comment 25: Line 64 – “It is also concluded…” Who is concluding or where the conclusion is presented is not articulated – referenced but not articulated.
Thank you for your correction. We have changed the text.
Lines 71-74
Brodaty et al. [11] concluded that the different interventions provided have a significant effect on the caregiver health, may reduce their psychological morbidity and help people with dementia to stay at home longer
Comment 26: Line 69 – “It is observed that support groups effect caregivers…” referenced but the source of the observation is not articulated, nor if the type of effect – e.g. positive impact on burden.
Thank you for your correction. Corrected previously (section 18 in the reviewer's review).
Lines 76-78
Support groups are seen to benefit caregivers and, so special emphasis needs to be placed on a better understanding of the effectiveness of support groups in the future.
Comment 27: Line 71 – “In a later review…” there is no review specified within the text prior to this statement to refer back to.
Thank you for your correction. We have changed the text.
Line 78-80
Jensen et al. [14] conclude in their systematic review that educational programs have a moderate positive effect on caregiver burden and a minor positive effect on depression.
Comment 28: Line 121-2 – two sentences beginning ‘firstly’ / ‘first’. This looks like an error and the meaning is unclear.
Thank you for your correction. We have changed the text.
Line 143
All studies had to meet the selection criteria. Firstly, an author developed the se-lection by reading the titles and abstract. Next, the selection was made by reading the title and abstract. Subsequently, the full content of the selected articles was obtained.
Comment 29: Line 290-91 – the meaning of the text in parentheses “(effectiveness will…” is not clear
Thank you for your correction. This phrase means that the longer the duration and frequency of the intervention, the better the results will be. We have changed the text.
Lines 318-319
The number of sessions per week and duration of the intervention (The effectiveness of the intervention will be better the longer and more frequently it is performed).
Comment 30: Line 325 – “Embarrassment is mainly to blame for…” more formal terminology could be more fitting here
Thank you for your correction. We have changed the text.
Lines 357-359
Shame is mainly to blame for the fact that this type of intervention is not very dominant or accepted by caregivers [12] and that few studies are potentially suitable.
Comment 31: Line 347-348 – I cannot make sense of this sentence.
Lines 381-383
Thank you for your correction. What we try to explain is the difficulty of identifying quality studies in relation to blinding due to the difficulty of performing correct blinding in the process of group assignment to each participant. We have changed the text.
A first limitation is the difficulty in finding studies of sufficient quality in relation to blinding of the intervention given the difficulty of carrying out this procedure in this type of study [88,89].
Comment 32. Supplementary tables: Table S3 Group therapy studies – mislabelled? Should this be S5?
Corrected.

Reviewer 3 Report
Comments and Suggestions for Authors
Thank you for the opportunity reviewing the manuscript titled “Interventions Effective in Decreasing Burden in Caregivers of Persons with Dementia: A Meta-analysis”. It is worth investigating the type of intervention that is the most appropriate to improve the health of the primary caregiver in patients with dementia.
The authors should be congratulated for their great effort in finalizing the high-quality manuscript.
I would like to provide below minor comments for the authors’ considerations:
1. The date of each database last search needs to be stated in the paper.
2. Would the authors report how to handle data from studies that included intention-to-treat?
Author Response
REVIEWER 3
Thank you for the time invested in the review process and for providing thoughtful feedback on our manuscript. Below, we have addressed each of your comments point-by- point and revised the manuscript accordingly. All the amendments in the revised manuscript are highlighted.
Comment 1. The date of each database last search needs to be stated in the paper.
The manuscript specifies the start and end dates of the search. We did not specify the end dates of the search for each database specifically because the searches were always performed in all the databases at the same time. In other words, when a new search was performed due to the incorporation of new information or due to the passing of time, it was carried out in all the databases at the same time. We added this remark in the manuscript for clarification of this point.
Lines 111-113:
The articles were obtained after conducting a systematic search with an initial date of September 30, 2019, to March 30, 2020. The different searches throughout this period were carried out in all databases at the same time.
Comment 2. Would the authors report how to handle data from studies that included intention-to-treat?
Thank you for your insightful observations. In the case of studies that provided an intention-to-treat analysis, we incorporated the information provided by the article for the calculation of the effect size. Generally, the information indicated was that obtained by the protocol analysis and that is what we used. However, we considered the percentage of dropouts as a moderating variable to explain the differences between effect sizes and did not find a statistically significant association. What we did detect was that there was an association between the percentage of losses and the year of publication, being higher in the first years.

Reviewer 4 Report
Comments and Suggestions for Authors
Methods: The search strategy is somewhat unclear, particularly in relation to search terms. The objectives are rather diverse and thus I feel would have required separate search strategies. Whilst I can see why the authors specifically included the Zarit scale in their search it maybe have been a limiting factor. Did the authors test their search strategy prior to running it?
The focus is very much on "burden" as an outcome (even though we have moved on from this concept and focus on support as it only focuses on negative aspects of caring) - yet the inclusion criteria specifies "interventions to improve caregiver burden" - does that mean interventions that worsened caregiver burden were excluded? The main objective of this review focused on improved health for dementia patients, yet the inclusion criteria only include care giver variables. Therefore there is a disconnect between aims/ objectives and methods and as a result affects the rest of the paper, please clarify what the main objective/ population was. However, given that the abstract and PICO state care givers as the main population, I am assuming that this is a typographical error (introduction, line 83).
The Prisma flowchart is somewhat confusing under the "identification" heading.
I cannot comment on the statistical analysis as that is not my area of expertise.
Discussion/ conclusion: It would be worthwhile to highlight the role of the nurse here further, given the journal's target audience.
Comments on the Quality of English Language
Some minor grammatical/ typographical errors throughout.
Author Response
REVIEWER 4
Thank you for the time invested in the review process and for providing thoughtful feedback on our manuscript. Below, we have addressed each of your comments point-by- point and revised the manuscript accordingly. All the amendments in the revised manuscript are highlighted. We have also tried to explain the aspects that you considered doubtful. After revising the manuscript according to your suggestions, we feel that its quality has been substantially improved. We hope you consider how we addressed the comments positively, and we look forward to hearing your decision.
Comment 1: Methods: The search strategy is somewhat unclear, particularly in relation to search terms.
Thank you for your comment. The literature search strategy is framed within the standards of systematic reviews in fact the search strategy has been approved and published in PROSPERO following the indicated standards. We chose a set of general keywords to obtain the set of articles related to those terms and then determined by inclusion and exclusion criteria which articles fit the PICO strategy. Obviously other much more specific keywords could be selected but from our point of view they could leave out interesting articles. This was a point of debate among the authors of the article and a consensus was reached. Other reviews in this field shared several of these keywords.
Comment 2: The objectives are rather diverse and thus I feel would have required separate search strategies. Whilst I can see why the authors specifically included the Zarit scale in their search it maybe have been a limiting factor. Did the authors test their search strategy prior to running it?
Thank you for your comment. The reviewer is right that there may be several objectives that touch on different aspects, although from our point of view the secondary objectives are inherent to the main objective. One question we asked ourselves was whether the type of instrument used could really affect the magnitude of the effect, since the diversity of instruments used was very large and, on the other hand, we were interested in the role of nurses versus other types of caregivers in this type of disease. Therefore, for us, the proposed objectives could be answered with the search carried out. It will be possible to disaggregate it in future lines of research. It is true that future research could focus on a specific aspect studied in this meta-analysis.
In order to carry out the search, we started with a set of articles that allowed us to outline the search. From these articles it was found that there were other instruments besides the Zarit scale that assessed caregiver health. These instruments were related to the Zarit scale, and some were improved and modified versions of the original instruments or new ones were constructed from them. The instruments used were: Caregiver Burden Index (CBI) [18], the Caregiver Strain Scale [19], the Impact of Caregiver Scale [20], the Memory and Behavior Problems Checklist (MBP) [21], Memory and Behavior Problem Checklist (BI) [22] and the Revised Memory and Behavior Problem Checklist (RMBPC) [23]. Lines 152-157.
Comment 3: The focus is very much on "burden" as an outcome (even though we have moved on from this concept and focus on support as it only focuses on negative aspects of caring) - yet the inclusion criteria specifies "interventions to improve caregiver burden" - does that mean interventions that worsened caregiver burden were excluded?
Thank you for your comment. Study groups have been established by type of intervention to improve caregiver overload. For each of the types of intervention according to previous authors, the study has indicated which ones have some type of benefit for the caregiver from the point of view of overload. Therefore, those interventions with the smallest effect size will be those that do not improve the caregiver's burden.
Comment 4: The main objective of this review focused on improved health for dementia patients, yet the inclusion criteria only include care giver variables. Therefore there is a disconnect between aims/ objectives and methods and as a result affects the rest of the paper, please clarify what the main objective/ population was. However, given that the abstract and PICO state care givers as the main population, I am assuming that this is a typographical error (introduction, line 83).
Thank you very much for your comment, indeed, it is a typographical error. We correct.
Line 92-94:
The main objective of the proposed review was which type of intervention is the most appropriate to improve the health of the primary caregiver in patients with dementia and which factors were important in explaining the heterogeneity.
Comment 5: The Prisma flowchart is somewhat confusing under the "identification" heading.
We have updated and adapted the flowchart to the new standards indicated by the Assistant Editor. We hope that the changes help the reader of the content.
Page, M. J.; McKenzie, J. E.; Bossuyt, P. M.; Boutron, I.; Hoffmann, T. C.; Mulrow, C. D.; Shamseer, L.; Tetzlaff, J. M.; Akl, E. A.; Brennan, S. E.; Chou, R.; Glanville, J.; Grimshaw, J. M.; Hróbjartsson, A.; Lalu, M. M.; Li, T.; Loder, E. W.; Mayo-Wilson, E.; McDonald, S.; McGuinness, L. A.; Stewart, L. A.; Thomas, J.; Tricco, A. C.; Welch, V. A.; Whiting, P.; Moher, D. The PRISMA 2020 Statement: An Updated Guideline for Reporting Systematic Reviews. BMJ 2021, n71. https://doi.org/10.1136/bmj.n71
http://prisma-statement.org/
http://www.prisma-statement.org/PRISMAStatement/FlowDiagram
Comment 6: I cannot comment on the statistical analysis as that is not my area of expertise.
Thank you for your sincerity. A statistician has been included among the authors to try to ensure the quality of the statistical methods.
Comment 7: Discussion/ conclusion: It would be worthwhile to highlight the role of the nurse here further, given the journal's target audience.
We appreciate your suggestion, which is indeed true. We have modified the paragraph where the role of nursing in this field was indicated.
Lines 415-424
The role of the nurse is essential when interventions are carried out to reduce the stress of the main caregiver of people with dementia. The nurse participates in many types of interventions and multidisciplinary groups. This study has detected that the presence of nurses in these multidisciplinary groups improves the effect of the intervention on caregivers when it is carried out in the form of workshops, although this does not imply that the role of nurses is not important in the rest of the interventions because the results were not statistically significant. The closeness and continuity of care that nurses provide is a historically recognized fact that may be related to this positive result. On the other hand, the presence of nurses is identified as a moderating variable that will help in future studies to detect other factors that influence the improvement of the results of the interventions.
